# An overseas business paradox: Are Japanese general contractors risk takers?

**Taichi Mutoh**[1], **Koji Kotani**[2,3,6,7]*, **Makoto Kakinaka**[4,5,6]

**1** Myanmar Office, International Marketing and Business Development Division, Taisei Corporation, Yangon, Myanmar, **2** School of Economics and Management, Kochi University of Technology, Kochi, Japan, **3** Research Institute for Future Design, Kochi University of Technology, Kochi, Japan, **4** Graduate School for International Development and Cooperation, Hiroshima University, Hiroshima, Japan, **5** Network for Education and Research on Peace and Sustainability (NERPS), Hiroshima University, Hiroshima, Japan, **6** College of Business, Rikkyo University, Tokyo, Japan, **7** Urban Institute, Kyushu University, Fukuoka, Japan

* kojikotani757@gmail.com

**Data Availability Statement:** Data is available as a Supporting Information file.

**Funding:** The first author of the manuscript (Taichi Mutoh) belongs to a private company of Taisei corporation. However, Taisei corporation and the

## Abstract

Japanese industries have struggled with stagnation after the collapse of bubble economy in the 1990s. Such a financial crisis has led to overseas business expansion of Japanese industries. This study empirically examines Japanese general contractors' overseas operations over the post-bubble period in relation to their financial status. The result shows that general contractors facing financial distress expand overseas business aggressively, when the domestic market shrinks. This result is opposite to conventional wisdom that stronger entities expand their territories of operations, thus "overseas business paradox." However, it can be considered a new scenario of industries' evolution when a country's economy matures.

## Introduction

There has been a hot debate of how firms are internationalized in global markets, and there are several theoretical studies to explain the process [1–3]. Johanson and Vahlne [1] introduce the Uppsala model to explain how a firm can be internationalized as a process. They argue that "establishment chain" through ad-hoc exporting, knowledge building and learning as well as "psychic distance" are important determinants for firms to be successfully internationalized. Johanson et. al. [2–4] further update the model by considering recent changes of business environment, and suggest that how a firm can be an insider in international business networks is crucial, while psychic distance has been less important in recent years. Thus, a firm is required to overcome the liability of outsiders to be an insider of international business networks, when the firm seeks to expand overseas operations.

In Japan, overseas construction operations have become active after the collapse of bubble economy in the early 1990s, although overseas operations are still considered difficult due to the uncertainties, complexities, and risks associated with differences in business cultures and practices. These facts have been frequently reported in various reports and articles written in Japanese. Japanese construction firms have responded to global competition by looking for

funders do not play any role in research activities concerning this manuscript, such as study design, data collection, analysis, decisions to publish and/ or preparation. The funders provided support with us in the form of salaries for authors of TM (Taichi Mutoh), KK (Koji Kotani) and MK (Makoto Kakinaka), but did not have any additional role.

**Competing interests:** We do not have any conflict of interests concerning this research. While the first author, Taichi Mutoh, belongs to a private company of Taisei corporation, it does not alter our adherence to all PLoS ONE policies on sharing data and materials.

new business opportunities in international markets beyond traditional domestic markets [5]. Technological superiority and financial capacity have contributed to the success of Japanese general contractors in international markets [6, 7]. Strategic alliances with Japanese manufacturers through massive foreign direct investment and Japan's construction aid have also facilitated market penetration of Japanese general contractors. The Japanese government has played some important roles in promoting Japanese general contractors in international markets by fostering technological and financial capacity [6]. Moreover, demand shrinkage for construction in the domestic market after the bubble period of the late 1980s has encouraged Japanese general contractors to engage in overseas business. Japanese general contractors still keep the share of overseas sales at the low level due to their conservative business behavior against project risks. We will illustrate this through summary statistics in later sections.

The cost of financing is one of the most important factors for Japanese general contractors to determine their overseas business expansion. Since the financing cost of a general contractor generally reflects the evaluation on its current and expected future performances in credit markets, general contractors with high financial status have the advantageous position in terms of the project cost, so that they could be expected to engage in overseas business in a more aggressive manner. However, general contractors with low financial status appear to implement overseas operations more aggressively than those with high financial status. Thus, we empirically study the determinants for the location mix of Japanese general contractors that go overseas in relation to their financial status, and seek to connect our results with the internationalization theory.

There are several empirical studies that analyze foreign direct investment (FDI) by multinational enterprises and show that FDI is driven by the possible exploitation of firm-specific advantages in various forms, such as ownership, location, and internalization [8–10]. More relevantly to this paper, a large number of works have examined locational determinants of FDI for multinational enterprises with an eye on various aspects, such as labor cost and quality, transportation and communication infrastructure, government policy, and industrial agglomeration, at the regional or national level. See some literature [11–23]. Among them, some works study location choices of FDI or overseas operations for Japanese investors [24–32].

Most of these empirical studies on overseas business expansion address manufacturers of a country during its high economic-growth period, and do not consider the relationship between firms' financial status (the cost of financing) and overseas operation. It should also be noticed that the construction industry differs from others, since general contractors are not entities that directly engage in FDI, and they receive orders of overseas projects from firms (typically manufacturers) which make a decision of direct investment. However, they have played a significant role in promoting economic growth for developing and developed countries since they are in charge of constructing hard and large-scale infrastructures for manufactures and countries.

Despite its importance, to the best of our knowledge, no empirical works study locational determinants of overseas business for general contractors. There are some studies on the internationalization of the construction industry of a high-economic growth period in some major countries [5–7, 33]. In addition, several studies have theoretically discussed an analytical framework of international entry decisions for construction firms in the field of decision theory [5, 34–38]. However, they do not empirically characterize the regional or spatial aspects of international business operations. Furthermore, few studies on overseas operations consider firms' financial status as well as the case of a country whose economy reaches maturity or even shrinks. Given this paucity, we examine overseas business activities of Japanese general contractors by incorporating their financial status into the analysis, and provide important implications about organizational behavior and development policy. In particular, the novelty of

our research lies in deriving a possible future scenario of industries in international business especially for a country whose economy reaches maturity. This research contributes to the theoretical models introduced [1–3], because a case of "matured countries" has never been considered in these models. We consider Japan as such a case, and the implication of our research is more valuable as many other countries are expected to follow the same type of paths in the near future Japan has been experiencing with respect to population and economic growth.

Our empirical analysis finds that general contractors facing significant financial distress are likely to expand their overseas business in a more aggressive manner. Irrespective of the measurements we use for the degree of internationalization (overseas operations) as a dependent variable, we confirm that the same qualitative conclusion holds. At first, this appears to be in sharp contrast to the conventional wisdom that advantageous firms with good financial status expand their overseas business. However, our paradoxical result can be meaningfully interpreted, when considering how Japanese business environment evolves over time. We call this result an "overseas business paradox" suggesting some possible future scenario of industries' evolution in a matured country.

After the collapse of the bubble economy in the early 1990s, the Japanese domestic construction market has shrunk due to the long-run economic distress with the reduction of public spending. Accordingly, many construction firms come to be recognized as "zombies" in the sense of Caballero et. al. [39], which need constant bailouts for their operation. The Japanese government provided domestic commercial banks with a series of bailouts in the post-bubble period, so-called "Japanese convoy system." While the general contractors had borrowed large amount of credits from the banks at that time, the bailouts made by the government had never been directly given to the general contractors. Consequently, however, a series of bailouts given by the government to the banks could be considered to have saved the general contractors as well. In this type of situations, our results suggest that general contractors without sound financial status are forced to receive orders of risky projects abroad for their survival, and otherwise would be forced to exit from the market.

In other words, the general contractors with low financial status can be considered those who are out of the profitable business network in Japanese "shrinking" construction industry. Therefore, they seek to be an insider of a new business network that may be in construction markets abroad. This story is in line with the updated Uppsala model [2]. The lesson from our paradoxical result could apply not only for the construction industry in Japan but also for some other industries in developed and emerging countries whose economy is expected to mature. As domestic markets become mature or shrunk, which is often observed in developed countries and may be experiential in developing countries in the near future, firms struggling with the high financing cost in a credit market may be forced to take higher risks and to expand their overseas business more aggressively.

## Construction industry in Japan

### Construction business

Construction business in Japan stands for the business industry, which consists of firms, called a contractor, making contracts on various building, architectural, and civil works provided under the Construction Business Act. The Act classifies the construction business into 28 types, and contractors are required to obtain license from either the Minister of Land, Infrastructure, Transport and Tourism or Prefectural Governors, depending on their business type. The Construction Business Act defines 28 kinds of business types, (1) general civil engineering, (2) general building, (3) carpentry, (4) plastering, (5) scaffolding, earthwork, and concrete, (6) masonry, (7) roofing, (8) electrical, (9) plumbing, (10) tile, brick, and block, (11) steel

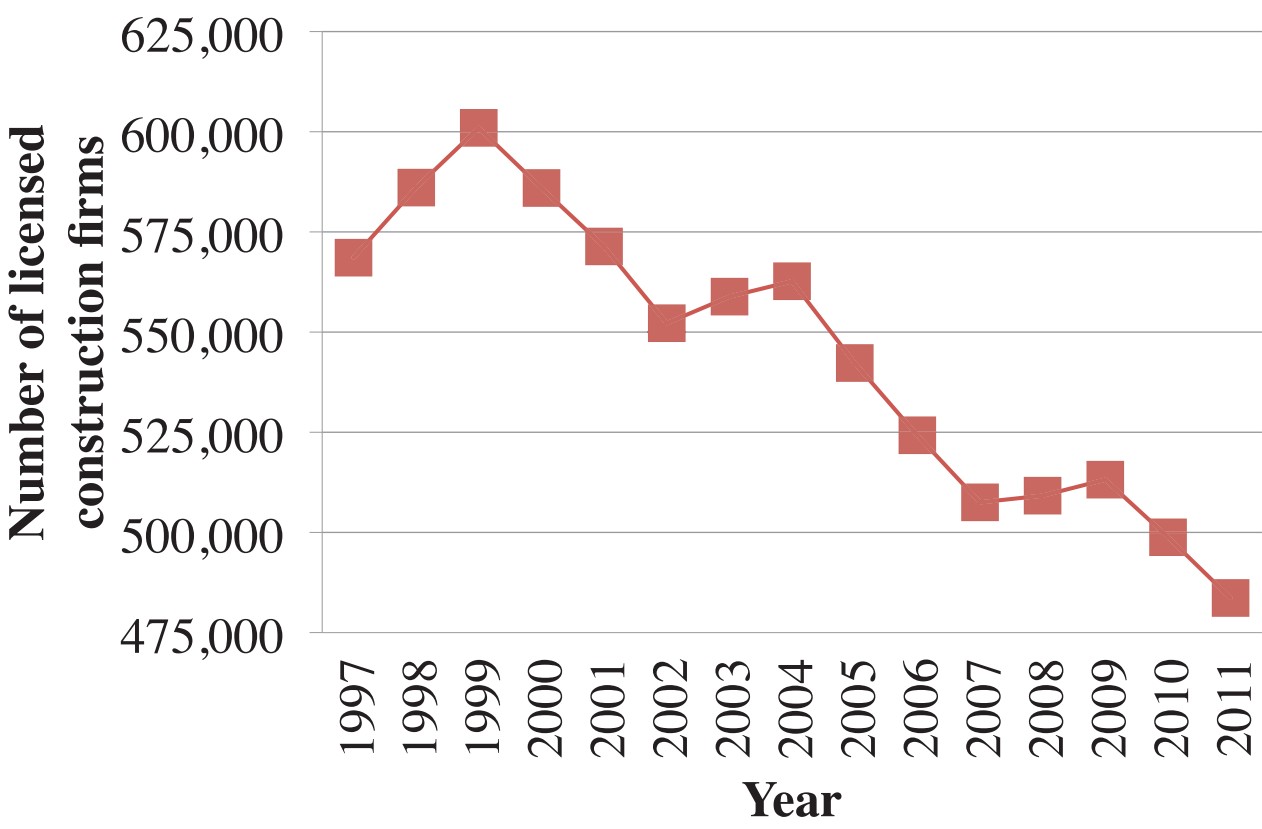

**Fig 1. Transition of the number of licensed construction firms.**

structure, (12) reinforcement steel, (13) paving, (14) dredging, (15) sheet metal, (16) glazing, (17) painting, (18) waterproofing, (19) interior finishing, (20) machine and equipment installation, (21) heat insulation, (22) telecommunication, (23) landscaping and gardening, (24) well drilling, (25) fittings, (26) water and sewerage facility, (27) fire protection facilities, and (28) sanitation facilities. Contractors are composed of main contractors, which contract a mega project (e.g., construction of large-scale airport, road network, dam, and skyscrapers), and subcontractors and sub-subcontractors, which contract parts of projects (e.g., carpentry, plumbing, and painting) with main contractors.

The number of contractors (construction firms) has been in a downward trend due mainly to economic distress and cuts in public spending on construction. According to the Ministry of Land, Infrastructure, Transport and Tourism, the number of contractors has declined by 15% from around 569000 in 1997 to around 484000 in 2011 (Fig 1). Among 28 types of business construction, over 30% of licensed firms have licenses of general building, scaffolding, earthwork and concrete, and general civil engineering. On the other hand, only less than 1% of licensed firms are given licenses of well drilling and sanitation facility. Another remark is that the number of construction firms holding only one license out of twenty eight is halved almost equally with the number of those obtaining multiple licenses. Table 1 illustrates the distribution of construction firms by the business scale as of 2011. Out of the whole construction industry, 98.8% of the firms are classified as medium- and small-sized enterprises with the capital amount of 100 million yen or less, and only 1.2% of the firms are classified as large-sized enterprises with the capital of 100 million yen or more. This implies that small-sized firms dominate the construction industry.

**Table 1. Distribution of contractors by capital, March 2011.**

| Amount of capital | Number of contractors | Proportions |
|---|---|---|
| Less than 5 million yen | 220828 | 45.7% |
| 5—10 million yen | 66462 | 13.7% |
| 10—100 million yen | 190683 | 39.4% |
| 100—1000 million yen | 4282 | 0.9% |
| 1—10 billion yen | 1027 | 0.2% |
| Over 10 billion yen | 357 | 0.1% |
| Total | 483639 | 100.0% |

Source: Ministry of Land, Infrastructure and Tourism.

The business formation in the Japanese construction industry is characterized as a "layered pyramid structure." A main contractor (general contractor) contracts the project with an employer (owner of the project) and takes the responsibility for the entire construction management to complete the project. It also issues subcontracts with special contractors and material suppliers, depending on the necessity and prompt timing to carry out the project efficiently. If needed, the subcontractors and the material suppliers issue further subcontracts with other construction-related firms. The formation of such a layered pyramid structure is more significant for large projects. In a megaproject, the number of subcontracts to be issued by the main contractor to subcontractors often exceeds over a few hundred. A megaproject is typically defined as a large-scaled investment with the amount of more than one billion US dollars. The responsibilities for contractual performance are basically fulfilled between the parties. Thus, the owner of the project is not in the position to intervene any contractual issues between the main contractor and its subcontractors. This logic remains valid to the lower-level contracts and it is usually used to risk avoidance to each layer.

## General contractors

This subsection describes the roles in the construction industry. In Japan, a business style of the layered pyramid structure has been playing an important role in the construction field for a long time. General contractors normally engage in contracts of civil or building projects in lump sum with their employers or owners and play a role as a main contractor to be responsible for the completion of the projects. The Japanese general contractors are at the top of the layered pyramid structure in a construction industry and receive the orders as a main contractor, not as a subcontractor. Among them, the five largest general contractors, Kajima, Obayashi, Shimizu, Taisei, and Takenaka, are particularly called a "super general contractor," which form the nucleus of the construction industry in Japan. Takenaka is not included in our sample, since it is not listed in the stock exchange market.

The construction industry has expanded with a number of general contractors due to the rising demand for construction during the rapid and stable economic growth period after World War II. Reconstruction in infrastructure and preparation for the 1964 Tokyo Olympic game are considered a remarkable event during the post-war period for not only the construction industry but the entire Japanese economy. However, after the collapse of the bubble economy with a sharp decline of asset prices in the early 1990s, many contractors, including general contractors, have struggled with the downturn in construction demands from private sectors and with the reduction in public investments associated with structural policy reforms. In fact, many contractors went into bankruptcy or kept alive under the assistance of financial

institutes, such as debt waiver, during the late 1990s and the early 2000s. The construction industry has recently attained an increase in sales, since there was an unexpected demand increase for the recovery, reconstruction, and nuclear related works as a result of massive earthquake in the Tohoku region in March 2011. The upward trend can be anticipated for several years due to the additional and increasing demands as well as new governmental policy to expand government expenditure. These contractors who could survive by the relief were usually forced to execute radical management reforms, leading them to be more shrunk and conservative. Such problematic firms could be observed particularly in the middle-scaled contractors or smaller.

Since Japanese general contractors generally rely heavily on the domestic construction market, they have a significant tendency that the share of domestic sales dominates that of offshore market sales, unlike foreign contractors, such as Vinci and Bouryguos in France, Hochtief in Germany, Skanska in Sweden, and Bechtel in the US, whose sales shares in overseas business are large. Table 2 shows the worldwide rankings up to the top 20 general contractors in terms of sales in 2006 and 2010, taken by Engineering News-Record (ENR) that provides information for the construction industry worldwide.

The share of overseas sales for the large-sized Japanese general contractors is around 10%, which is much lower than major foreign contractors. The low level of overseas operations for Japanese general contractors can be explained by the argument that most of them could maintain their business in the domestic market and thus they do not take a risk of foreign projects aggressively. Recent trend of demand shrinkage for construction after the bubble period may encourage Japanese general contractors to receive foreign projects, although most general contractors still keep the share of overseas sales at the low level due to conservative business behavior. Table 2 also presents that major Chinese general contractors record the low ratio of overseas sales. However, differently from Japan, this is due mainly to the fact that Chinese economy has drastically been growing in the recent decades. In addition, it should be noted that the ratio of overseas sales for most of major Chinese general contractors has increased, although their domestic share is still high. This clearly shows that major Chinese general contractors make the importance on both domestic and international markets.

## Overseas business expansion of general contractors

The business expansion of Japanese general contractors to overseas markets started with the Seoul-Inchon railway construction in Korea (Joseon Dynasty) in 1897-1900, which was undertaken by Kajima Corporation, one of the major general contractors. Okura-Gumi, a precursor firm of Taisei Corporation, currently being one of the major general contractors in Japan, established its London branch in 1874. This might be the first overseas business base among Japanese firms. However, the business formation of Okura-Gumi was not related to construction, but was a kind of trading firm dealing with machineries and military weapons. During the pre-war period, Japanese general contractors expanded overseas business operations mainly for infrastructure development in Japan's territorial region. After World War II, Japanese general contractors restarted to go abroad, Korea and Asian countries. At this stage, they were involved in overseas business expansion in a passive way under the war reparations. Since the 1960s, they have gradually transferred their overseas business associated with government foreign policy toward commercial based business. Their overseas business was further expanded, along with the international construction boom, in the 1970s due mainly to the demand from the middle-east countries backed up by oil money. The amount of the order position in the overseas market was about 20 billion yen in the early 1970s, and it achieved a sudden surge up to 500 billion yen during the decade.

**Table 2. Worldwide ranking in sales among construction firms.**

| | Name of firm | Country | Sales | Offshore sales | Offshore sales ratio |
|---|---|---|---|---|---|
| | Year 2006 | | | | |
| 1 | Vinci | France | 32699 | 11065 | 33.8% |
| 2 | Bouyguos | France | 24960 | 9576 | 38.4% |
| 3 | Chinal Highway Engineering | China | 21296 | 658 | 3.1% |
| 4 | Hochtief | Germany | 19795 | 17599 | 88.9% |
| 5 | Grupo ACS | Spain | 18527 | 3004 | 16.2% |
| 6 | China Railway Construction | China | 17327 | 415 | 2.4% |
| 7 | China State Construction Engineering | China | 16147 | 2956 | 18.3% |
| 8 | Skanska | Sweden | 15722 | 12347 | 78.5% |
| 9 | Bechtel | USA | 15367 | 8931 | 58.1% |
| 10 | China Communication Construction | China | 14734 | 3381 | 22.9% |
| 11 | Taisei | Japan | 14176 | 2069 | 14.6% |
| 12 | Kajima | Japan | 13981 | 2151 | 15.4% |
| 13 | Eiffage | France | 13970 | 2010 | 14.4% |
| 14 | Strabag | Austria | 13502 | 10799 | 80.0% |
| 15 | Shimizu | Japan | 12673 | 1343 | 10.6% |
| 16 | Obayashi | Japan | 12462 | 1779 | 14.3% |
| 17 | Fcc. Fomento | Spain | 11894 | 2155 | 18.1% |
| 18 | China Metalhurgical | China | 11628 | 907 | 7.8% |
| 19 | Takenaka | Japan | 11293 | 1649 | 14.6% |
| 20 | Fluor | USA | 11274 | 6339 | 56.2% |
| | Year 2010 | | | | |
| 1 | China Railway Construction | China | 76206 | 3424 | 4.5% |
| 2 | China Railway Group | China | 73012 | 3158 | 4.3% |
| 3 | China State Construction Engineering | China | 48868 | 4871 | 10.0% |
| 4 | Vinci | France | 45111 | 16557 | 36.7% |
| 5 | China Communication Construction | China | 40418 | 7134 | 17.7% |
| 6 | Bouyguos | France | 30671 | 12432 | 40.5% |
| 7 | China Metalhurgical | China | 29905 | 1514 | 5.1% |
| 8 | Hochtief | Germany | 28979 | 27424 | 94.6% |
| 9 | Grupo ACS | Spain | 20631 | 6562 | 31.8% |
| 10 | Bechtel | USA | 19714 | 12500 | 63.4% |
| 11 | Leighten Holdings | Australia | 18510 | 3648 | 19.7% |
| 12 | Eiffage | France | 17729 | 2853 | 16.1% |
| 13 | Fluor | USA | 17194 | 11565 | 67.3% |
| 14 | Fcc. Fomento | Spain | 16059 | 7457 | 46.4% |
| 15 | Sinohydro | China | 15883 | 4010 | 25.2% |
| 16 | Skanska | Sweden | 14635 | 11632 | 79.5% |
| 17 | Shimizu | Japan | 14403 | 1162 | 8.1% |
| 18 | Kajima | Japan | 14394 | 2106 | 14.6% |
| 19 | Obayashi | Japan | 13675 | 1916 | 14.0% |
| 20 | Shanghai Construction | China | 13005 | 1654 | 12.7% |

Source: Engineering news-record (ENR).

Sales and offshore sales are in terms of million US dollars.

The next boost emerged in the early 1980s when the amount of the order position rose from around 500 billion yen to the level of 1 trillion yen. The main reasons include overseas expansions of Japanese manufactures through foreign direct investment (FDI) and infrastructure development through official development assistance (ODA) in developing countries, especially in Asia. The development of the construction industry in Asia during the 1980s can be characterized by three trends: (1) more participation of private sectors in infrastructure projects, (2) vertical integration in the packaging of construction projects, and (3) foreign participation in domestic construction, and these trends can be attributed to the globalization and deregulation of markets [6]. The success of Japanese general contractors can be attributed to technological superiority, financial capacity, and formation of strategic alliances with local governments and firms [6]. In particular, ODA has been carried out continuously, contributing to Japanese general contractors' order position despite the significant decline in domestic demand. Moreover, the Japanese government has supported overseas contracting through informal pressures and coordination with the Sogo Shosha or private trading companies [33]. Strassmann [33] emphasizes the role of government support with finance during the period after the 1980s, particularly for Japanese, French, and Italian firms. In general, government supports take the form of export credits, tax preferences, trade promotion, tied foreign aid, and negotiating countertrade. Raftery et. al. [6] also present important roles in promoting Japanese general contractors by fostering technological and financial capacity.

## Empirical analysis

This section conducts empirical analysis to discuss the role of financial conditions in making the location choice of overseas business operations for Japanese general contractors. We first provide an explanation of the methodology and data in our estimation. After showing several preliminary results, we present the results of our estimation and their implications. Recall that we analyze the location mix of Japanese general contractors that go overseas, which is different from the investment decision. As mentioned earlier, the general contractors are not entities that directly engage in FDI, other types of investments and aids such as ODA, and they simply receive the orders of overseas projects from firms (typically manufacturers) which make a decision of direct investment. In this sense, our analysis fundamentally differs from the previous researches that analyze the location of FDI.

### Methodology and data

The following empirical model is estimated:

$$\mathrm{OP}_{i,j,t} = \alpha_0 + \alpha_1 \mathrm{FIS}_{j,t-1} + \alpha_2 \mathrm{CSIZE}_{j,t-1} + \sum_k \beta_k z_{k,i,t-1} + \epsilon_{i,j,t},$$

where $\mathrm{OP}_{i,j,t}$ is the measure of overseas business operations of contractor $j$ in country $i$ at year $t$, $\mathrm{FIS}_{j,t}$ is the measure of financial status for contractor $j$ at year $t$, $\mathrm{CSIZE}_{j,t}$ is the measure of firm size for contractor $j$ at year $t$, $z_{k,i,t}$ is variable $k$ of country-specific factors in country $i$ at year $t$ and $\epsilon_{i,j,\,t}$ is an error term with standard properties. In addition to $\mathrm{FIS}_{j,t}$ as our main independent variable, we include firm size $\mathrm{CSIZE}_{j,t}$, which is measured by the log of the asset of contractor $j$, as a contractor-specific factor since it is well acknowledged that large-sized firms tend to be in an advantageous position due to the economies of scale and scope. This study uses the lag variables for all independent variables to avoid possible endogeneity problems as in previous literature.

There are many studies on internationalization and globalization of enterprises over the past decades, but how to measure the degree of internationalization of a firm appears to remain an unsolved issue. Among various measures, foreign sales or revenues may be a

meaningful first-order indicator of firm's involvement in overseas business operations [40]. In this study, the model takes each of the following two measures of overseas operations as a dependent variable, $OP_{i,j,t}$, for robustness check of our empirical results. The first dependent variable (OCC) is a count variable which takes the number of orders of the overseas projects received by contractor $j$ in country $i$. The second (OCA) is the log of one plus the total real value of the orders of the overseas project received by contractor $j$ in country $i$ in terms of the US dollar, which is adjusted by the US Consumer Price Index (CPI).

In our analysis, financial status is regarded as the overall credibility or evaluation on each contractor in a credit market. When a contractor receives an order of the overseas project, it generally needs to obtain the credit from banks for the deposit associated with the order. The contractor with high credibility in a credit market tends to be offered bank loan with the low interest rate. In contrast, for the contractor with less credibility in a credit market, banks tend to offer loan with the high interest rate due to the high risk premium. Thus, financial status, or the credibility in a credit market, would influence the financing costs for each contractor. Our analysis captures financial status for each contractor by using the measure of market-based evaluation. For this purpose, we first construct the hypothetical interest payment:

$$R_{j,t}^* = r_t^S D_{j,t}^S + r_t^L D_{j,t}^L,$$

where $r_t^S$ is the short-term prime rate at year $t$, $r_t^L$ is the average long-term prime rate over the past three years from the year $t$, $D_{j,t}^S$ is the average of the short-term debt of contractor $j$ in year $t-1$ and year $t$, and $D_{j,t}^L$ is the average of the long-term debt of contractor $j$ in year $t-1$ and year $t$. The short-term debt is comprised of any debt that is due within one year, and the long-term debt is comprised of other debts that are due in a greater than 12-month period. This hypothetical interest payment can be considered the interest payment that applies for the contractor with the highest credibility in a credit market, taking $D_{j,t}^S$ and $D_{j,t}^L$ as given. Then we construct the measure of the financial status for contractor $j$ (FIS):

$$\text{FIS}_{j,t} = r_{j,t} - r_{j,t}^* = \frac{R_{j,t} - R_{j,t}^*}{D_{j,t}},$$

which is equivalent to the gap between the actual and hypothetical interest rates, where $R_{j,t}$ is the actual interest payment for contractor $j$ at year $t$.

The value of $\text{FIS}_{j,t}$ reflects general contractor's financial status in a credit market. This particularly reflects the credit rating, which is an evaluation of the credit worthiness of a debtor, including profitability and risk in current and future periods. The evaluation is made by a credit rating agency of the debtor's ability to pay back the debt and the likelihood of default. If a general contractor entails the high credibility in a credit market, the actual interest payment is close to the hypothetical one, so that the financial status $\text{FIS}_{j,t}$ is relatively low. If a general contractor has financial problems, currently or in the future, due mainly to the expectation of low profitability, then the lender requests high risk premium, so that the actual interest payment is higher than the hypothetical one. In this case, the financial status $\text{FIS}_{j,t}$ is relatively high.

Concerning the country-specific factors to be expected to affect the decision of overseas business operations, we include variables related to official development assistance from Japan to country $i$ ($ODA_{i,t}$) and foreign direct investment inflow from Japan to country $i$ ($FDI_{i,t}$), which are measured by the log of one plus real ODA from Japan to country $i$ and the log of one plus real FDI inflow from Japan to country $i$, respectively. The overseas activities of general contractors are generally associated with the projects financed through ODA by public sectors or FDI by private enterprises, as mentioned in Raftery et. al. and Ofori [6, 7], so that ODA and

FDI are expected to enhance general contractors' overseas expansion. Japanese general contractors generally pay attention to financing sources of projects. Based on the financing sources, projects can mainly be classified into ODA-loan, ODA-grant, and self-funded projects. Ideally, the model should take into account the financing sources of projects. However, we do not consider such financing sources in our analysis due to the absence of such detailed data.

In addition, we include the size of the economy of country $i$ ($ESIZE_{i,t}$), which is measured by the log of real GDP, to capture how the economic size affects general contractors' overseas activities. More business opportunities for construction firms may exist in a large country. However, large economies have already established hard infrastructure with the less demand for construction. Thus, the impact of the economic size on the overseas activities depends on which one dominates the other. The model further includes the income difference between Japan and country $i$ ($INCM_{i,t}$), which is measured by real per capita GDP of Japan minus that of country $i$, to capture how per capita income or skill difference affects the overseas business activities. Moreover, the measure of political stability in country $i$ ($POLIT_{i,t}$) is included in the model to evaluate the impact of political risk.

Furthermore, the model includes the degree of Japanese general contractors' concentration in country $i$ ($CONC_{i,t}$), which is defined by the Hirshmann-Herfindahl Index (HHI) for each country and each year:

$$CONC_{i,t} = \sum_j h_{i,j,t}^2,$$

where $h_{i,j,t}^2$ is the relative exposure of general contractor $j$ in country $i$ at year $t$, which is calculated by the amount of received orders by contractor $j$ in country $i$ divided by the total amount of received orders by all Japanese general contractors. The degree of the concentration provides general contractors with a signal of how Japanese firms have operated in their business. If many of general contractors have already been under operation, they might believe that their own operation could also obtain the profit successfully. In this case, the impact of the concentration on the overseas business activities could be negative.

The data set of order position records published by the Overseas Construction Association of Japan (OCAJI) is used to construct the panel data of the two measures of overseas business operations ($OP_{i,j,t}$) during the sample period from 1998 to 2010. This data set of OCAJI shows information about all overseas projects received by 65 membership companies (including most Japanese general contractors) with the details of the projects, such as the received contractors, the amount of orders received, country (location where to implement), fund source, and executing agency in the country. There exist overseas projects received by non-membership contractors of OCAJI, like relatively small construction firms. However, most cases are covered in the data set, since firms with overseas business typically become a membership of OCAJI partly to collect information related to their business. In other words, it can be considered that the results of our analysis may not change even if we include the data of overseas projects received by non-membership contractors. Another possible problem is that our dataset contains the information of orders of overseas projects but not the information of whether projects were completed fully or only partially completed. These factors could ultimately influence the actual rate the contractor faces. Although we agree that disbursement-based information, including fully/partially completion of projects, is more reliable, we use the order-based information in our analysis due to the data limitation.

The contractor-specific data of financial position, such as asset, short-term and long-term debts, and interest payment, is obtained from Kaisya-Shikiho (Japan Company Handbook) and Datastream. Concerning the country-specific information, the data of bilateral real official

development assistance is taken from Creditor Reporting System (CRS), maintained by the Development Assistance Committee (DAC) of the Organization for Economic Cooperation and Development (OECD), containing information on international aid and activity-level aid. In particular, we use the committed amounts of bilateral ODA.

Although the disbursed amounts would be more appropriate, they are not available only for some donors, as DAC mentioned in user's guide. The data of nominal FDI flows and nominal trade (import plus exports) flows are taken from the International Direct Investment Statistics of the OECD and the Direction of Trade Statistics of the IMF (DOTS-IMF), respectively. To construct real FDI and trade flows, we divide nominal flows by the US GDP deflator, which is obtained from the World Development Indicators (WDI) of World Bank. As other country-specific variables, the data of real GDP and real per capita GDP are taken from the WDI, and the measure of political stability is taken from political risk rating of International Country Risk Guide (ICRG). Moreover, the short-term and long-term prime rates are taken from the Bank of Japan.

Our unbalanced panel data set consists of 16145 observations with 36 contractors and 72 countries during the sample period from 1998 to 2010, due to incomplete data of some country-specific and contractor-specific variables. Tables 3 and 4 present the lists of general contractors and countries in the sample used in our empirical analysis, respectively. As mentioned earlier, the two measures of overseas business operations (OCC and OCA) are used as a dependent variable.

The OCC is a count variable capturing the number of orders of overseas contracts for each general contractor and each country. Many studies, including those on FDI location choice, have applied count data models [19, 26, 30, 41–43]. This study applies Poisson models and negative binomial models (NBMs). The assumed equality of the conditional mean and variance can be considered the major shortcoming of the Poisson regression models. Among several alternatives, the most common is negative binomial models (NBMs). The NBM is an extension of the Poisson regression model by introducing an individual, unobserved effect into the conditional mean.

One possible methodological problem is that since each general contractor has many countries where it has no operations, the dependent variable contains many zero counts, so that the distribution of the OCC data is skewed to the right and contains a large proportion of zeros (i.e., excess zeros). The data of no operations provides relevant information, since the independent variable containing zeros could help explain the reason why general contractors do not receive any orders of contracts in some specific countries. To deal with the problem related to the distributional characteristics, this study applies a zero-inflated Poisson and negative binomial regression models. The excess of zeroes is incorporated in zero-inflated models, which is a finite mixture model, where one distribution is considered as a degenerate process with a unit point mass at zero [44]. The model allows for excess zeros in count models under the assumption that the population is characterized by two groups: one group whose counts are generated by the Poisson or negative binomial model, and another group (absolute zero group) that have zero probability of a count greater than zero. Observed zero counts could come from either group. The likelihood of being in either group is estimated using a logit model [45]. The zero-inflated negative binomial model allows overdispersion through the splitting process that models the outcomes.

The OCA captures the total real value of the orders of overseas projects received by each contractor in each country. For this dependent variable, we apply ordinary least squares (OLS) for the estimation. However, similar to our count models, the model estimation may suffer from a zero problem, which is often discussed in the international trade literature, since OCA variable contains a significant portion of zero values. In our data set, for each contractor, there

**Table 3. List of general contractors.**

|  | Name of general contractor |
|---|---|
| 1 | Ando Corporation |
| 2 | Aoki Corporation |
| 3 | Daiho Corporation |
| 4 | Fujita Corporation |
| 5 | Fukuda Corporation |
| 6 | Hazama Corporation |
| 7 | Hitachi Plant Technologies |
| 8 | JDC Corporation |
| 9 | Kajima Corporation |
| 10 | Kandenko |
| 11 | Kinden Corporation |
| 12 | Kitano Construction |
| 13 | Kumagai Gumi |
| 14 | Maeda Corporation |
| 15 | Nakano Kubota Construction |
| 16 | Nippon Road |
| 17 | Nishimatsu Construction |
| 18 | Obayashi Corporation |
| 19 | Ohki Corporation |
| 20 | Okumura Corporation |
| 21 | P.S. Mitsubishi Construction |
| 22 | Penta Ocean Construction |
| 23 | Sato Kogyo |
| 24 | Shimizu Corporation |
| 25 | Sumitomo Mitsui Construction |
| 26 | Taisei Corporation |
| 27 | Takenaka Civil Engineering & Construction |
| 28 | Tekken Corporation |
| 29 | Toa Corporation |
| 30 | Tobishima Corporation |
| 31 | Toda Corporation |
| 32 | Tokura Construction |
| 33 | Tokyu Construction |
| 34 | Toyo Construction |
| 35 | Wakachiku Construction |
| 36 | Zenitaka Corporation |

are many countries in which it does not receive any orders of contracts, as mentioned in the previous discussion. This kind of zero-contract amounts is considered as a corner solution outcome in the context of economic theory, where typical OLS estimation may not be appropriate. To mitigate this issue, we apply the Tobit model by using the log of one plus the total real value of the orders of overseas projects as the dependent variable. In addition, following the work of Santos Silva and Tenreyro [46], we also apply the Poisson pseudo-maximum likelihood (PPML) estimations. The PPML estimation method can be applied to the levels of the total real value of the orders of overseas projects, rather than their log forms, to estimate directly the nonlinear form of the model. All estimated models include the year and contractor dummies to control for the year- and contractor-specific effects.

**Table 4. List of countries.**

| | Code | Name | | Code | Name |
|---|---|---|---|---|---|
| 1 | AGO | Angola | 37 | KWT | Kuwait |
| 2 | ARE | United Arab Emirates | 38 | LBN | Lebanon |
| 3 | ARG | Argentina | 39 | LKA | Sri Lanka |
| 4 | AZE | Azerbaijan | 40 | MAR | Morocco |
| 5 | BFA | Burkina Faso | 41 | MDG | Madagascar |
| 6 | BGD | Bangladesh | 42 | MEX | Mexico |
| 7 | BHR | Bahrain | 43 | MLI | Mali |
| 8 | BRA | Brazil | 44 | MNG | Mongolia |
| 9 | BRN | Brunei | 45 | MWI | Malawi |
| 10 | CHL | Chile | 46 | MYS | Malaysia |
| 11 | CHN | China | 47 | NER | Niger |
| 12 | CIV | Cote d'Ivoire | 48 | NGA | Nigeria |
| 13 | CMR | Cameroon | 49 | NIC | Nicaragua |
| 14 | COL | Colombia | 50 | OMN | Oman |
| 15 | CRI | Costa Rica | 51 | PAK | Pakistan |
| 16 | DOM | Dominican Republic | 52 | PAN | Panama |
| 17 | DZA | Algeria | 53 | PER | Peru |
| 18 | ECU | Ecuador | 54 | PHL | Philippines |
| 19 | EGY | Egypt | 55 | PNG | Papua New Guinea |
| 20 | ETH | Ethiopia | 56 | PRY | Paraguay |
| 21 | GAB | Gabon | 57 | SAU | Saudi Arabia |
| 22 | GHA | Ghana | 58 | SEN | Senegal |
| 23 | GIN | Guinea | 59 | SGP | Singapore |
| 24 | GMB | The Gambia | 60 | SLE | Sierra Leone |
| 25 | GNB | Guinea-Bissau | 61 | SLV | El Salvador |
| 26 | GUY | Guyana | 62 | SUR | Suriname |
| 27 | HKG | Hong Kong SAR, China | 63 | SYR | Syrian Arab Republic |
| 28 | HND | Honduras | 64 | THA | Thailand |
| 29 | HRV | Croatia | 65 | TUN | Tunisia |
| 30 | IDN | Indonesia | 66 | TUR | Turkey |
| 31 | IND | India | 67 | TZA | Tanzania |
| 32 | IRN | Iran | 68 | UGA | Uganda |
| 33 | IRQ | Iraq | 69 | VNM | Vietnam |
| 34 | JAM | Jamaica | 70 | YEM | Yemen |
| 35 | JOR | Jordan | 71 | ZAF | South Africa |
| 36 | KEN | Kenya | 72 | ZMB | Zambia |

## Some preliminaries

This subsection first examines the characteristics of dependent and independent variables used in the estimation. Then we briefly discuss the relationship between overseas business operations and financial status in a credit market. For the better understanding, we divide full sample into the two subsamples, depending on the value of financial status measures (FIS). The first corresponds to the subsample where the value of FIS is below its median (0.0101), and the second corresponds to the subsample where the value of FIS is above its median. Table 5 shows the summary statistics of our main variables. It is observed that the means of overseas business operations (OCC and OCA) for the subsample of low FIS are larger than

**Table 5. Summary statistics.**

| Variable | Observation | Mean | Std. dev. | Min | Max |
|---|---|---|---|---|---|
| Full sample | | | | | |
| OCC | 16145 | 0.636 | 3.805 | 0.000 | 116.000 |
| OCA | 16145 | 0.230 | 0.868 | 0.000 | 7.130 |
| FIS | 16145 | 0.011 | 0.012 | −0.008 | 0.162 |
| CSIZE | 16145 | 12.656 | 1.052 | 10.416 | 14.899 |
| ODA | 16145 | 3.757 | 2.135 | 0.000 | 8.579 |
| FDI | 16145 | 2.144 | 2.190 | 0.000 | 8.794 |
| ESIZE | 16145 | 25.158 | 1.741 | 21.257 | 29.743 |
| INCM | 16145 | 2.055 | 1.139 | −0.859 | 4.043 |
| POLIT | 16145 | 64.296 | 8.951 | 35.500 | 90.000 |
| CONC | 16145 | 0.720 | 0.329 | 0.081 | 1.000 |
| Subsample of low FIS (FIS < Median) | | | | | |
| OCC | 8046 | 0.711 | 3.708 | 0.000 | 69.000 |
| OCA | 8046 | 0.288 | 0.987 | 0.000 | 6.376 |
| FIS | 8046 | 0.005 | 0.004 | −0.008 | 0.010 |
| CSIZE | 8046 | 12.848 | 1.175 | 10.416 | 14.899 |
| DA | 8046 | 3.739 | 2.137 | 0.000 | 8.579 |
| FDI | 8046 | 2.062 | 2.885 | 0.000 | 8.794 |
| ESIZE | 8046 | 25.126 | 1.738 | 21.257 | 29.743 |
| INCM | 8046 | 2.073 | 1.137 | −0.859 | 4.043 |
| POLIT | 8046 | 64.296 | 8.937 | 35.500 | 90.000 |
| CONC | 8046 | 0.723 | 0.329 | 0.081 | 1.000 |
| Subsample of high FIS (FIS > Median) | | | | | |
| OCC | 8099 | 0.561 | 3.898 | 0.000 | 116.000 |
| OCA | 8099 | 0.172 | 0.728 | 0.000 | 7.130 |
| FIS | 8099 | 0.018 | 0.013 | 0.010 | 0.162 |
| CSIZE | 8099 | 12.465 | 0.873 | 10.457 | 14.865 |
| ODA | 8099 | 3.775 | 2.133 | 0.000 | 8.579 |
| FDI | 8099 | 2.226 | 2.933 | 0.000 | 8.794 |
| ESIZE | 8099 | 25.287 | 1.743 | 21.257 | 29.743 |
| INCM | 8099 | 2.038 | 1.141 | −0.859 | 4.043 |
| POLIT | 8099 | 64.296 | 8.966 | 35.500 | 90.000 |
| CONC | 8099 | 0.718 | 0.329 | 0.081 | 1.000 |

those for the subsample of high FIS, which implies that credible general contractors tend to engage more in overseas business operations compared to less credible general contractors. On the other hand, the means of other variables show less significant differences between the two subsamples.

Table 6 the correlation matrix for the full sample and the two subsamples. First, the size of the contractor (CSIZE) and the economic size of a country (ESIZE) are positively correlated with overseas business operations (OCC and OCA). Large-sized general contractors tend to engage more in overseas business expansion, and general contractors tend to expand their business toward relatively large-sized countries. Second, bilateral ODA and FDI flows (ODA and FDI) are also positively correlated with overseas operations. Overseas business expansion might be promoted through foreign aid and FDI from Japan. Third, the difference in per capita income (INCM) and political stability (POLIT) are negatively and positively associated

**Table 6. Correlation matrix of main variables.**

|  | OCC | OCA | FIS | CSIZE | ODA | FDI | ESIZE | INCM | POLIT | CONC |
|---|---|---|---|---|---|---|---|---|---|---|
| Full sample | | | | | | | | | | |
| OCC | 1.00 | | | | | | | | | |
| OCA | 0.65 | 1.00 | | | | | | | | |
| FIS | 0.01 | −0.04 | 1.00 | | | | | | | |
| CSIZE | 0.15 | 0.24 | −0.22 | 1.00 | | | | | | |
| ODA | 0.11 | 0.06 | 0.00 | −0.01 | 1.00 | | | | | |
| FDI | 0.24 | 0.29 | 0.02 | −0.03 | 0.14 | 1.00 | | | | |
| ESIZE | 0.16 | 0.18 | 0.01 | −0.02 | 0.24 | 0.84 | 1.00 | | | |
| INCM | −0.06 | −0.13 | 0.00 | 0.02 | 0.53 | −0.61 | −0.39 | 1.00 | | |
| POLIT | 0.06 | 0.14 | 0.00 | 0.00 | −0.40 | 0.37 | 0.07 | −0.61 | 1.00 | |
| CONC | −0.24 | −0.33 | 0.00 | 0.00 | −0.21 | −0.64 | −0.43 | 0.22 | −0.23 | 1.00 |
| Subsample (low FIS): FIS < median (0.010) | | | | | | | | | | |
| OCC | 1.00 | | | | | | | | | |
| OCA | 0.68 | 1.00 | | | | | | | | |
| FIS | −0.02 | −0.04 | 1.00 | | | | | | | |
| CSIZE | 0.21 | 0.28 | −0.30 | 1.00 | | | | | | |
| ODA | 0.12 | 0.07 | 0.00 | 0.01 | 1.00 | | | | | |
| FDI | 0.27 | 0.31 | −0.01 | 0.00 | 0.14 | 1.00 | | | | |
| ESIZE | 0.17 | 0.20 | 0.00 | 0.00 | 0.23 | 0.73 | 1.00 | | | |
| INCM | −0.08 | −0.16 | 0.00 | 0.00 | 0.53 | −0.41 | −0.39 | 1.00 | | |
| POLIT | 0.07 | 0.16 | 0.01 | 0.00 | −0.38 | 0.29 | 0.07 | −0.60 | 1.00 | |
| CONC | −0.29 | −0.37 | 0.01 | 0.00 | −0.22 | −0.60 | −0.43 | 0.22 | −0.24 | 1.00 |
| Subsample (high FIS): FIS > median (0.010) | | | | | | | | | | |
| OCC | 1.00 | | | | | | | | | |
| OCA | 0.63 | 1.00 | | | | | | | | |
| FIS | 0.05 | 0.02 | 1.00 | | | | | | | |
| CSIZE | 0.08 | 0.14 | −0.13 | 1.00 | | | | | | |
| ODA | 0.09 | 0.06 | 0.00 | −0.04 | 1.00 | | | | | |
| FDI | 0.22 | 0.27 | 0.01 | −0.07 | 0.14 | 1.00 | | | | |
| ESIZE | 0.16 | 0.16 | −0.01 | −0.05 | 0.24 | 0.74 | 1.00 | | | |
| INCM | −0.05 | −0.10 | 0.01 | 0.04 | 0.54 | −0.41 | −0.39 | 1.00 | | |
| POLIT | 0.05 | 0.11 | 0.00 | 0.01 | −0.41 | 0.29 | 0.08 | −0.62 | 1.00 | |
| CONC | −0.20 | −0.29 | 0.00 | −0.01 | −0.20 | −0.59 | −0.43 | 0.23 | −0.23 | 1.00 |

with overseas business operations, respectively. These appear to suggest that general contractors are likely to expand their overseas business toward the countries with the relatively high income and political stability. Fourth, the concentration measure (CONC) is negatively correlated with overseas business operations, so that general contractors tend to expand their business toward the countries where other Japanese contractors have already been under operations. Fifth, more relevantly to the objective of this study, financial status in a credit market (FIS) appears to be uncorrelated with overseas business operations.

The comparison of the correlation matrix for the two subsamples shows that the correlations between the size of the contractor (CSIZE) and overseas business operations (OCC and OCA) for the subsample of low FIS are larger than those for the subsample of high FIS. The positive association of contractor's size with overseas business operations is more significant for the group of credible general contractors. In addition, the comparison also presents that

the negative correlations between the concentration measure (CONC) and overseas business operations (OCC and OCA) for the subsample of low FIS are more significant than those for the subsample of high FIS. The negative association of the concentration measure with overseas business operations is more significant for the group of credible general contractors. On the other hand, the correlations between overseas business operations (OCC and OCA) and other variables show less significant differences between the two subsamples of low FIS and high FIS.

Table 7 presents the average of several variables related to overseas business operations and financial status over the sample period. It is easily observed that large-sized general contractors, such as Kajima, Obayashi, Shimizu, and Taisei, have received a large amount of contracts in foreign countries. At the same time, their spread between the actual and hypothetical interest rates is relatively small so that their financial status is advantageous in credit market. On the other hand, the relatively small-sized contractors have received a small amount of contracts, and their financial status is relatively low. However, once we adjust the amount of contracts by using the size of general contractors (total asset), the simple analysis in Table 7 may fail to show a clear relationship between financial status and overseas business operations, as in correlation matrix of Table 6. To carefully discuss how general contractors in our sample decide their overseas business in relation to their financial status in a credit market, we conduct empirical analysis by applying some econometric methods in the next subsection.

## Results

This subsection shows the results of our estimations to evaluate general contractors' location choice of overseas business operations and discuss how financial status affects their decision. Tables 8 to 10 show the "full sample" results of our empirical models with OCC and OCA as the dependent variable. In addition, as in the previous subsection, we also divide full sample into the two subsamples, depending on the value of FIS. In Tables 8 to 10, columns of low FIS show the estimated results for the subsample where the value of FIS is below its median (0.0101), while those of high FIS present the estimated results for the subsample where the value of FIS is above its median. By doing so, we can evaluate the interaction effects of financial status and other control variables on overseas business operations.

**Financial status in a credit market.** The result consistently shows that irrespective of the subsamples of low and high FIS, the coefficients on financial status (FIS) are significantly positive for all measures of overseas business operations, except for high FIS in the Poisson and negative binomial parts of the zero-inflated Poisson and negative binomial models and in the PPML model. It should be noted that the setup of the logit model in the zero-inflated models is to predict the probability of being no operations, so that a negative coefficient on an independent variable implies a positive relationship between the probability of operations and the independent variable. Since the high value of FIS implies the low evaluation in a credit market due mainly to the low profitability or the high default risk, the result suggests that less credible general contractors tend to expand overseas business operations by receiving orders of overseas projects. Given the argument that overseas business operations are risky in general, less credible general contractors tend to take a higher risk than highly credible ones. Moreover, Tables 8 to 10 show that the estimated coefficients for the subsample of low FIS is larger than those for the subsample of high FIS. This implies that the measures of overseas business operations increase with a rise in FIS in a concave manner, i.e., the effect of financial status in credit markets on overseas business operations would decrease as financial status worsens.

Several possible explanations can be considered on this result related to the positive association between financial status and overseas business operations. The first factor originates from

**Table 7. Numbers of countries and contracts, amount of contracts and financial status (average over the period 1998-2010).**

| Contractor | Number of countries | Number of contracts | Amount of contracts | Total asset | Contract to asset ratio | Financial status |
|---|---|---|---|---|---|---|
| | | | (A) | (B) | (A)/(B) | |
| Ando Corporation | 3.5 | 17.8 | 6261 | 186130 | 0.034 | 0.95 |
| Aoki Corporation | 1.1 | 3.8 | 1598 | 534900 | 0.003 | 0.06 |
| Daiho Corporation | 2.2 | 3.8 | 7289 | 140388 | 0.052 | 0.89 |
| Fujita Corporation | 9.6 | 104.3 | 20592 | 502337 | 0.041 | 1.61 |
| Fukuda Corporation | 0.4 | 2.2 | 120 | 164081 | 0.001 | 1.01 |
| Hazama Corporation | 10.2 | 47.3 | 18013 | 162088 | 0.111 | 2.52 |
| Hitachi Plant Technologies | 1.2 | 6.8 | 3809 | 230044 | 0.017 | 0.83 |
| JDC Corporation | 0.2 | 0.3 | 355 | 420363 | 0.001 | 0.80 |
| Kajima Corporation | 11.9 | 117.8 | 117682 | 2061538 | 0.057 | 0.70 |
| Kandenko | 1.6 | 3.4 | 402 | 380015 | 0.001 | 1.42 |
| Kinden Corporation | 5.7 | 47.6 | 6248 | 514322 | 0.012 | 2.60 |
| Kitano Construction | 3.3 | 4.5 | 3008 | 70624 | 0.043 | 1.61 |
| Kumagai Gumi | 6.5 | 29.3 | 26024 | 697031 | 0.037 | 1.23 |
| Maeda Corporation | 4.9 | 22.4 | 25801 | 566771 | 0.046 | 1.00 |
| Nakano Kubota Construction | 1.3 | 10.8 | 2402 | 79722 | 0.030 | 0.93 |
| Nippon Road | 1.2 | 2.6 | 492 | 140266 | 0.004 | 0.11 |
| Nishimatsu Construction | 7.3 | 35.5 | 47158 | 709949 | 0.066 | 0.13 |
| Obayashi Corporation | 9.6 | 77.5 | 76490 | 1953846 | 0.039 | 0.06 |
| Ohki Corporation | 0.6 | 1.1 | 396 | 107059 | 0.004 | 0.67 |
| Okumura Corporation | 0.5 | 0.8 | 1890 | 405434 | 0.005 | 1.83 |
| P.S. Mitsubishi Construction | 0.2 | 0.2 | 210 | 96736 | 0.002 | 2.85 |
| Penta Ocean Construction | 7.9 | 23.9 | 66766 | 426938 | 0.156 | 1.30 |
| Sato Kogyo | 3.0 | 19.9 | 19942 | 716534 | 0.028 | 0.92 |
| Shimizu Corporation | 14.5 | 83.5 | 88181 | 1907692 | 0.046 | 0.34 |
| Sumitomo Mitsui Construction | 7.9 | 125.0 | 28578 | 393761 | 0.073 | 3.73 |
| Taisei Corporation | 15.0 | 120.6 | 102338 | 1961538 | 0.052 | 0.48 |
| Takenaka Civil Engineering & Construction | 0.2 | 0.5 | 152 | 80767 | 0.002 | 3.04 |
| Tekken Corporation | 1.0 | 1.6 | 2759 | 216301 | 0.013 | 0.69 |
| Toa Corporation | 4.4 | 5.8 | 14371 | 253144 | 0.057 | 0.91 |
| Tobishima Corporation | 4.4 | 10.9 | 4641 | 252035 | 0.018 | 0.61 |
| Toda Corporation | 5.6 | 45.8 | 8651 | 648316 | 0.013 | 1.30 |
| Tokura Construction | 2.3 | 3.1 | 2070 | 36579 | 0.057 | 1.00 |
| Tokyu Construction | 2.3 | 10.1 | 4427 | 175838 | 0.025 | 1.37 |
| Toyo Construction | 1.8 | 13.7 | 5346 | 208378 | 0.026 | 1.71 |
| Wakachiku Construction | 1.3 | 1.6 | 1050 | 128909 | 0.008 | 1.42 |
| Zenitaka Corporation | 2.5 | 6.9 | 2020 | 240747 | 0.008 | 1.08 |

Japan's experience of a long-term macroeconomic stagnation after the collapse of the bubble economy in the early 1990s. The construction industry in Japan generally depends on public infrastructure projects, such as roads, bridges, and highways construction projects. However, the long-term economic distress, along with some other factors such as aging society with increased social security burden, has caused local and central governments to face a drastic increase in public debts. Due to this budget problem, the governments have been unable to

**Table 8. Locational choice of international operations.**

| Variables | OCC | | | | | | OCA | | | | | |
|---|---|---|---|---|---|---|---|---|---|---|---|---|
| | POISSON | | | NBREG | | | OLS | | | TOBIT | | |
| | Full sample | Low FIS | High FIS | Full sample | Low FIS | High FIS | Full sample | Low FIS | High FIS | Full sample | Low FIS | High FIS |
| FIS | 19.038*** | 105.353*** | 11.936 | 20.667*** | 91.679*** | 25.293*** | 1.790*** | 8.612*** | 1.120** | 22.084*** | 102.565*** | 24.912*** |
| | (4.986) | (21.811) | (9.395) | (6.857) | (23.073) | (10.220) | (0.530) | (3.021) | (0.572) | (8.380) | (32.086) | (11.994) |
| CSIZE | 0.500** | 1.483*** | 0.474*** | 1.010*** | 2.041*** | 1.061*** | 0.162*** | 0.205*** | 0.158*** | 1.876*** | 3.470*** | 2.001*** |
| | (0.183) | (0.546) | (0.218) | (0.223) | (0.545) | (0.268) | (0.031) | (0.075) | (0.036) | (0.272) | (0.844) | (0.324) |
| ODA | 0.092*** | 0.158*** | 0.015 | 0.108*** | 0.136*** | 0.085** | 0.015*** | 0.018*** | 0.010* | 0.164*** | 0.200*** | 0.107* |
| | (0.027) | (0.032) | (0.041) | (0.029) | (0.035) | (0.043) | (0.004) | (0.007) | (0.006) | (0.040) | (0.052) | (0.059) |
| FDI | 0.254*** | 0.201*** | 0.336*** | 0.265*** | 0.188*** | 0.370*** | 0.052*** | 0.051*** | 0.053*** | 0.284*** | 0.163*** | 0.447*** |
| | (0.030) | (0.033) | (0.051) | (0.023) | (0.025) | (0.039) | (0.004) | (0.007) | (0.005) | (0.035) | (0.043) | (0.057) |
| ESIZE | −0.042 | −0.165*** | 0.067 | −0.101** | −0.115** | −0.124* | −0.041*** | −0.040*** | −0.041*** | −0.231*** | −0.102 | −0.384*** |
| | (0.045) | (0.044) | (0.068) | (0.042) | (0.049) | (0.067) | (0.006) | (0.009) | (0.007) | (0.056) | (0.072) | (0.088) |
| INCM | −0.099 | −0.270*** | 0.093 | −0.225*** | −0.290*** | −0.131 | −0.032*** | −0.053*** | −0.010 | −0.338*** | −0.446*** | −0.174 |
| | (0.071) | (0.082) | (0.115) | (0.069) | (0.085) | (0.106) | (0.008) | (0.013) | (0.009) | (0.101) | (0.134) | (0.149) |
| POLIT | 0.001 | 0.002 | −0.001 | 0.000 | 0.008 | −0.007 | 0.002*** | 0.004*** | 0.001 | 0.013 | 0.028*** | −0.006 |
| | (0.005) | (0.006) | (0.009) | (0.006) | (0.007) | (0.010) | (0.001) | (0.001) | (0.001) | (0.009) | (0.011) | (0.013) |
| CONC | −3.695*** | −4.490*** | −2.888*** | −3.802*** | −4.358*** | −3.360*** | −0.649*** | −0.864*** | −0.438*** | −6.338*** | −6.832*** | −5.622*** |
| | (0.281) | (0.323) | (0.417) | (0.160) | (0.193) | (0.240) | (0.026) | (0.042) | (0.029) | (0.244) | (0.306) | (0.374) |
| Constant | −7.886*** | −14.834*** | −11.121*** | −11.017*** | −22.474*** | −11.363*** | −0.341 | −0.939 | −0.541 | −19.617*** | −40.704*** | −17.319*** |
| | (1.978) | (6.240) | (2.711) | (2.872) | (6.254) | (3.672) | (0.478) | (0.881) | (0.557) | (3.556) | (9.806) | (4.454) |
| Obs | 16145 | 8046 | 8099 | 16145 | 8046 | 8099 | 16145 | 8046 | 8099 | 16145 | 8046 | 8099 |
| R-squared | 0.607 | 0.660 | 0.587 | - | - | - | 0.217 | 0.258 | 0.163 | 0.262 | 0.281 | 0.246 |

***, ** and * are significant at the 1%, 5% and 10% levels, respectively.

All models include contract and year dummies.

Robust standard errors are in parentheses.

keep a high level of public spending and have been enforced to cut public spending, particularly on infrastructure development. Public opinion against the unnecessary infrastructure has also supported this policy.

Such an environment with weak business sentiment associated with a long-term economic distress has reduced the demand for construction from public institutions as well as private enterprises in domestic markets. This would reduce firms' profitability and increase their business risk in the construction industry, including general contractors. To mitigate this issue, some general contractors have been encouraged to seek for the opportunities of their business expansion in foreign countries with the expectation of higher profit. This tendency may be amplified more significantly for general contractors struggling with low profitability and high default risk, which is assumed to be captured by our measure of financial status (FIS). The result can be considered consistent with the Uppsala model updated by Johanson et. al. [2–4] in the sense that the general contractors with low financial status seek to be an insider of new business networks abroad to be profitable again when they are out of profitable business network in domestic markets. That is, less credible general contractors (high FIS) are more likely to expand overseas business operations (high OP).

The second factor affecting the relationship between financial status and overseas business operations is related to the financing of infrastructure and industrial projects. General

**Table 9. Locational choice of international operations: Zero-inflated Poisson and negative binomial regressions.**

| Variables | Zero-inflated POISSON | | | | | | Zero-inflated NBREG | | | | | |
|---|---|---|---|---|---|---|---|---|---|---|---|---|
| | Full sample | | Low FIS | | High FIS | | Full sample | | Low FIS | | High FIS | |
| | POISSON | Zero inf | POISSON | Zero inf | POISSON | Zero inf | NBREG | Zero inf | NBREG | Zero inf | NBREG | Zero inf |
| FIS | 8.591** | −19.835*** | 62.715*** | −41.631*** | 1.754 | −29.779*** | 15.358*** | −14.571*** | 78.409*** | −37.018** | 10.597 | −21.342*** |
| | (4.168) | (3.847) | (23.483) | (13.331) | (7.271) | (5.162) | (5.910) | (4.882) | (24.936) | (16.314) | (8.808) | (6.658) |
| CSIZE | 0.015 | −0.964*** | −0.066 | −1.001*** | −0.099 | −0.754*** | 0.377 | −0.938*** | 0.849 | −0.970*** | 0.407 | −0.658*** |
| | (0.612) | (0.045) | (0.528) | (0.056) | (0.545) | (0.088) | (0.241) | (0.056) | (0.784) | (0.078) | (0.302) | (0.100) |
| ODA | 0.036 | −0.125*** | 0.064** | −0.160*** | −0.013 | −0.081** | 0.047 | −0.117*** | 0.065* | −0.149*** | −0.011 | −0.098* |
| | (0.025) | (0.025) | (0.029) | (0.035) | (0.039) | (0.036) | (0.031) | (0.033) | (0.039) | (0.046) | (0.053) | (0.051) |
| FDI | 0.187*** | −0.125*** | 0.194*** | −0.059** | 0.172*** | −0.211*** | 0.183*** | −0.111*** | 0.180*** | −0.042 | 0.212*** | −0.184*** |
| | (0.028) | (0.023) | (0.033) | (0.029) | (0.041) | (0.037) | (0.026) | (0.029) | (0.033) | (0.039) | (0.042) | (0.045) |
| ESIZE | 0.002 | 0.141*** | −0.146*** | 0.045 | 0.161*** | 0.255*** | −0.013 | 0.157*** | −0.130 | 0.029 | 0.120 | 0.286*** |
| | (0.041) | (0.036) | (0.050) | (0.049) | (0.057) | (0.054) | (0.062) | (0.055) | (0.088) | (0.089) | (0.076) | (0.069) |
| INCM | −0.096 | 0.195*** | −0.119 | 0.227** | −0.005 | 0.136 | −0.193*** | 0.127 | −0.208*** | 0.150 | 0.065 | 0.199 |
| | (0.070) | (0.067) | (0.082) | (0.092) | (0.117) | (0.101) | (0.092) | (0.097) | (0.106) | (0.127) | (0.171) | (0.155) |
| POLIT | −0.012** | −0.009* | −0.011* | −0.022*** | −0.014* | 0.005 | −0.008 | −0.009 | −0.009 | −0.024*** | −0.002 | 0.010 |
| | (0.005) | (0.005) | (0.006) | (0.007) | (0.008) | (0.008) | (0.006) | (0.007) | (0.007) | (0.009) | (0.010) | (0.011) |
| CONC | −0.959*** | 3.419*** | −1.472*** | 3.721*** | −0.550* | 3.072*** | −1.539*** | 3.266*** | −2.127*** | 3.458*** | −0.411 | 3.280*** |
| | (0.245) | (0.173) | (0.353) | (0.249) | (0.309) | (0.259) | (0.250) | (0.234) | (0.344) | (0.337) | (0.331) | (0.319) |
| Constant | 0.673*** | 10.488*** | 6.108 | 13.999*** | −2.657 | 4.628*** | −2.294 | 9.314*** | −4.325 | 13.794*** | −8.961*** | 1.358 |
| | (1.915) | (1.190) | (6.095) | (1.621) | (2.657) | (1.922) | (2.294) | (1.571) | (8.644) | (2.400) | (4.084) | (2.456) |
| Obs | 16145 | | 8046 | | 8099 | | 16145 | | 8046 | | 8099 | |
| Likelihood[1] | −8693.15 | | −4644.87 | | −3738.72 | | −6498.94 | | −3559.90 | | −2857.12 | |

***, ** and * are significant at the 1%, 5% and 10% levels, respectively.

All models include contract and year dummies.

Robust standard errors are in parentheses.

[1] "Likelihood" denotes the log-pseudolikelihood for each estimation.

contractors typically need to obtain credits from financial institutions when they implement an overseas project. The financing cost is crucial when a general contractor obtains credit in a credit market. Credible financial status enables the general contractor to obtain credits at the low financing cost and to implement the project at the low cost. Thus, credible general contractors have the advantage in competitive bids or more generally, the sealed bid process, which is often applied in construction contracts, since competitive bidding aims at implementing the project at the lowest costs by stimulating competition and by preventing favoritism. This argument implies that less credible general contractors (high FIS) are less likely to expand overseas business operations (low OP), in contrast to the discussion in the first factor.

The positive association between FIS and OP in our estimated results suggests that the first factor dominates the second, so that less credible general contractors (high FIS) are more likely to expand overseas business operations (high OP). We have interviewed several managers and executives that work in Japanese general contractors. When we explained the results of our analysis, they had thought for a while about whether our results are consistent with what they have experienced in the workplace with respect to overseas operations. They agreed that those firms which become unprofitable in domestic construction markets appear to more aggressively take overseas projects for survival. Since their profitability is low in domestic construction markets, these firms tend to receive low credit rating as well. Administrators of overseas

**Table 10. Locational choice of international operations: Poisson pseudo-maximum likelihood (PPML) estimation where the dependent variable is the total real value of the orders for the overseas projects.**

| Variables | Full sample | Low FIS | High FIS |
|---|---|---|---|
| FIS | 31.710*** | 104.401*** | 11.743 |
| | (9.907) | (41.755) | (11.407) |
| CSIZE | 0.801*** | 1.353* | 0.657** |
| | (0.230) | (0.801) | (0.300) |
| ODA | −0.004 | 0.045 | −0.119** |
| | (0.042) | (0.058) | (0.048) |
| FDI | −0.020 | −0.096*** | −0.282*** |
| | (0.034) | (0.034) | (0.062) |
| ESIZE | 0.082 | 0.139** | −0.181** |
| | (0.050) | (0.059) | (0.082) |
| INCM | −0.468*** | −0.629*** | 0.027 |
| | (0.162) | (0.223) | (0.142) |
| POLIT | 0.025*** | 0.022* | 0.033*** |
| | (0.010) | (0.013) | (0.013) |
| CONC | −4.310*** | −4.520*** | −3.368*** |
| | (0.337) | (0.416) | (0.531) |
| Constant | −13.667*** | −19.196 | −5.563 |
| | (3.116) | (11.743) | (4.239) |
| Obs | 16145 | 8046 | 8099 |
| *R*-squared | 0.283 | 0.304 | 0.497 |

***, ** and * are significant at the 1%, 5% and 10% levels, respectively.

All models include contract and year dummies.

Robust standard errors are in parentheses.

projects usually prefer to contract with financially strong general contractors in the bidding process. However, since overseas projects, mostly in developing countries, are perceived as high-risk projects and less profitable than domestic projects, financially strong general contractors seem to be reluctant to participate in competitive bidding or to set a very high price in competitive bidding. On the other hand, financially weak general contractors seem to be eager to take such high risk partly to keep their operations and employment. Our findings appear to be in sharp contrast to the argument of the world history showing that stronger entities have expanded their territory of operation. We call this paradoxical argument an "overseas business paradox." Since the early 1990s, the domestic construction market has shrunk due to the long-run economic distress with the reduction of public spending. In this situation, general contractors without sound financial status would be forced to receive orders of risky projects abroad for their survival, although their financing cost is relatively high. The lesson from our paradoxical argument could apply not only for the construction industry in Japan but also for some industries in developed and emerging countries. As domestic markets become mature or shrunk, which is often observed in developed countries and may be experiential in developing countries in the future, firms struggling with the high financing cost in a credit market may take high risks by expanding their overseas business. To verify this argument related to domestic and overseas projects, we need more careful analyses, including the comparison between domestic and overseas projects. However, due to the unavailability of the detailed data of

domestic and overseas projects, such analyses are difficult in our empirical framework. The careful examination will be left for future research.

Caballero et. al. [39] suggest that Japanese banks have been involved in sham loan restructurings which kept credit flowing to otherwise insolvent borrowers, which is called 'zombies.' Zombie firms have obtained subsidized credits from banks through various financial assistances, such as debt forgiveness, interest rate concessions, debt for equity swaps, the reduction in interest payments, and moratoriums on interest payments. By constructing several measures of zombieness based on the subsidized credits over the period from 1981 to 2002, they present that during the 1990s and the early 2000s, the zombie problem was more serious for non-manufacturing industries, particularly the construction industry, than for manufacturing industries. A possible reason for the cross-industrial differences includes the intensified global competition, where manufacturing firms could not be protected easily by their banks. Another reason may be that the construction and real estate industries had a significant negative impact of the collapse of asset prices, including land prices [39]. The zombie-related arguments imply that if banks had not provided subsidized loans, zombie contractors would have paid higher interest payments and thus have been characterized as the higher value of our financial status measure (FIS). In this case, the balance of the first and second factors, mentioned in the above discussions, determines how financial conditions would have influenced the location choice of overseas business operations for zombie contractors.

**Other control variables.**    Tables 8 to 10 also present the estimation results related to other control variables, CSIZE, ODA, FDI, ESIZE, CONC, INCM, and POLIT, all of which are expected to affect general contractors' location choice. Dividing full sample into the two subsamples of low FIS and high FIS allows us to verify the existence of the interaction effects of financial status and other control variables. The coefficients on the firm size (CSIZE), as another contractor-specific control variable, are significantly positive for all models in Tables 8 to 10 and are significantly negative for the logit part of the zero-inflated models in Table 9, which implies that large-sized general contractors tend to engage more in overseas business expansion. Possible justification for this result includes that large-sized general contractors implement projects in various fields of construction-related services so that they can comply with the requirement of projects' employers in foreign countries. In addition, the estimated coefficients on the firm size (CSIZE) for the subsample of low FIS is larger than those for the subsample of high FIS, which implies that the sensitivity of overseas business operations in response to the firm size is large for general contractors with high credibility in credit markets (low FIS).

Concerning country-specific control variables, the coefficients on official development assistance (ODA) and foreign direct investment (FDI) in Table 8 are significantly positive for most models, irrespective of the two subsamples of low FIS and high FIS. In addition, the coefficients on ODA and FDI in Table 9 are positive and negative for the Poisson (negative binomial) part and the logit part of the zero-inflated models, respectively. Bilateral foreign aid by Japanese government and foreign investment by Japanese firms, particularly Japanese manufacturers, would encourage general contractors to expand overseas business operations. It is well known that one of the main targets of Japan's foreign aid is to promote infrastructure development in recipient countries. One possible obstacle for Japanese general contractors to receive the contract order is that under the current regulation of ODA from Japan, the tender procedure is open for any nationalities if the bidder satisfies the criteria given by executing agencies in the host country, even though the fund comes from Japanese government. Such a circumstance causes Japanese firms to face the intense competition against international bidders, especially Chinese and Korean firms with the cost-related advantage. Another problem is the financing issue related to the fact that for most of infrastructure development, covering all

costs through ODA is almost impossible. Thus, Japanese firms are recommended to establish new business schemes, including operation after completion of the construction, and other alternative financing schemes, such as Public Private Partnership (PPP), where private business venture is often funded and operated through a partnership of the recipient government and private enterprises. However, some projects require advanced technology, and Japanese firms generally have the advantage in construction technology and experiences. Thus, some grant aid projects are the exceptions from the open tender system, so that only Japanese firms are eligible to implement these projects. The positive association of ODA with overseas business operations in our empirical analysis suggests the positive role of foreign aid from Japan in helping Japanese general contractors' expansion of their business to foreign countries, although the open tender system intensifies the competition with foreign contractors.

In addition to foreign aid from Japan, the positive association of FDI with overseas business operations implies that direct investment of Japanese firms is also one of the crucial factors for Japanese general contractors' behavior. It should be noticed that the party to engage in foreign investment is not contractors themselves, but manufacturers, such as automobiles, electrical parts, textile, and retail dealers. Foreign investment of Japanese manufacturers creates business opportunities to Japanese general contractors. When Japanese manufacturers set up new factories or facilities, they often order new construction to Japanese general contractors although they are free to choose non-Japanese firms. This is due mainly to the motivation to mitigate various risk factors, including the construction period and the quality of buildings, through the long-term reliance established between general contractors and manufacturers. In particular, the manufacturers that start business in a specific country without proper knowledge and information tend to order Japanese general contractors as a kind of inward security.

The comparison of the estimated coefficients on ODA and FDI for the two subsamples generally suggests that the coefficients on ODA for the subsample of low FIS are larger than those for the subsample of high FIS, while the coefficients on FDI for the subsample of low FIS are smaller than those for the subsample of high FIS. When ODA from Japan increases, general contractors with high credibility (low FIS) tend to expand their overseas business operations more aggressively, compared to general contractors with low credibility (high FIS). In contrast, when FDI from Japan increases, general contractors with low credibility (high FIS) tend to expand their overseas business operations more aggressively, compared to general contractors with high credibility (low FIS). These results illustrate that credible general contractors with advantageous financing costs tend to receive the orders of relatively less risky ODA-related projects, while less credible general contractors tend to receive the orders of projects financed through direct investment by Japanese firms.

For other country-specific control variables, the analysis presents that the size of economy (ESIZE) are negatively associated with overseas business operations, although some models show less clear or inconsistent results. This result partly supports that Japanese general contractors tend to expand their overseas business operations in small-sized countries. In addition, per capita income (INCM) is negatively associated with overseas business operations for the full sample and the subsample of low FIS, while there is no clear relationship between them for the subsample of high FIS. This implies that general contractors, particularly credible general contractors, are reluctant to expand their overseas business operations to low-income countries. Moreover, political stability (POLIT) is positively associated with overseas business operations particularly for the subsample of low FIS, although some models show insignificant results. This result suggests that credible general contractors are likely to pay more attention to political stability of the country when they expand their overseas business operations. Finally, the concentration measure (CONC) is negatively correlated with overseas business operations, so that Japanese general contractors are likely to expand their overseas business to the

countries where other general contractors have already been under operations. In other words, Japanese general contractors may be characterized as a follower of other successful firms in each country. The larger value of the estimated coefficients on CONC for the subsample of low FIS implies that this tendency would be more substantial for credible general contractors.

## Conclusion

Since the collapse of the bubble economy in the early 1990s, Japan has experienced a long-term economic distress, which has caused Japanese business society to emphasize the importance of overseas business expansion for their survival. The construction industry is no exception to this trend. Focusing on the role of market-based financial status in a credit market, this study has examined location choices of Japanese general contractors' overseas business expansion over the post-bubble period. The conventional wisdom suggests that firms with the high corporate performance take advantage of overseas business expansion. However, in sharp contrast to this argument, our results have shown clear evidence of the paradoxical argument, "overseas business paradox," i.e., general contractors facing financial distress tend to expand their overseas business in a more aggressive manner.

The result is in line with the Uppsala model in the sense that the general contractors with low financial status seek to be an insider of new business networks abroad to be profitable when they are out of profitable business network in domestic markets. The lesson from our paradoxical results could apply not only for the construction industry in Japan but also for some other industries in developed and emerging countries. In other words, our empirical finding is interpreted as a possible future scenario of industries' evolution when the economy of a single country matures. This type of economic maturities may be observed in developed countries and be experiential in some emerging countries in the near future. Then our results imply that less credible firms with low profitability and high default risk in domestic markets have stronger incentives of overseas business expansion for their survival. This result is quite inconsistent with what has happened in territory expansion of world history, i.e., stronger entities expand their territories. However, it is our belief that what we find in this paper could be considered a new path of how industries can evolve in globalized international business.

## Supporting information

**S1 File. Excel "overseasbusiness.xlsx" data file.** It contains all the necessary data to replicate the statistical and regression results presented in this paper.
(XLSX)

## Acknowledgments

The authors thank anonymous referees, Takahiro Akita, Hiroaki Miyamoto and Kenta Tanaka for their helpful comments, advice and supports.

## Author Contributions

**Conceptualization:** Taichi Mutoh, Koji Kotani, Makoto Kakinaka.

**Data curation:** Koji Kotani, Makoto Kakinaka.

**Formal analysis:** Taichi Mutoh.

**Investigation:** Taichi Mutoh.

**Methodology:** Koji Kotani, Makoto Kakinaka.

**Project administration:** Taichi Mutoh.

**Resources:** Makoto Kakinaka.

**Software:** Taichi Mutoh.

**Supervision:** Makoto Kakinaka.

**Validation:** Koji Kotani.

**Visualization:** Koji Kotani.

**Writing – original draft:** Taichi Mutoh, Koji Kotani.

**Writing – review & editing:** Koji Kotani.

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
