## [Decision Letter · Decision Letter 0]

27 May 2020

PONE-D-20-02677

An overseas business paradox: Are Japanese general contractors risk takers?

PLOS ONE

Dear Dr. Kotani,

Thank you for submitting your manuscript to PLOS ONE. After careful consideration, we feel that it has merit but does not fully meet PLOS ONE’s publication criteria as it currently stands. Therefore, we invite you to submit a revised version of the manuscript that addresses the points raised during the review process.

We look forward to receiving your revised manuscript.

Kind regards,

Petre Caraiani

Academic Editor

PLOS ONE

Journal Requirements:

'No role by funders'   

We note that one or more of the authors are employed by a commercial company: 'Taisei Corporation'.

Additional Editor Comments (if provided):

Please respond in writing and by modifying the paper accordingly to all the comments by the referee.

Reviewers' comments:

Reviewer's Responses to Questions

**Comments to the Author**

1. Is the manuscript technically sound, and do the data support the conclusions?

Reviewer #1: Yes

2. Has the statistical analysis been performed appropriately and rigorously? 

Reviewer #1: Yes

3. Have the authors made all data underlying the findings in their manuscript fully available?

Reviewer #1: Yes

4. Is the manuscript presented in an intelligible fashion and written in standard English?

Reviewer #1: Yes

5. Review Comments to the Author

Reviewer #1: This is an interesting analysis studying the overseas activity of Japanese general contractors. My comments are listed below:

1. p. 16: is there any data on whether projects were completed fully or only partially completed? Or if projects took longer than expected. Those factors could ultimately influence the actual rate the contractor faces.

2. p. 17: as you note, there are a large number of zeros in your data. Have you considered using a zero-modified Poisson distribution in your analysis?

3. What level of competition exists in bids? Why are some overseas projects willing to accept a bid from a higher risk contractor. Can you explain this process more.

4. Do you have data for these contractors on their domestic projects so that comparisons can be made on their actual interest rates for domestic versus overseas projects?

6. PLOS authors have the option to publish the peer review history of their article (what does this mean?). If published, this will include your full peer review and any attached files.

Reviewer #1: No

---

## [Decision Letter · Decision Letter 1]

20 Aug 2020

An overseas business paradox: Are Japanese general contractors risk takers?

PONE-D-20-02677R1

Dear Dr. Kotani,

We’re pleased to inform you that your manuscript has been judged scientifically suitable for publication and will be formally accepted for publication once it meets all outstanding technical requirements.

Kind regards,

Petre Caraiani

Academic Editor

PLOS ONE

Additional Editor Comments (optional):

Reviewers' comments:

Reviewer's Responses to Questions

**Comments to the Author**

1. If the authors have adequately addressed your comments raised in a previous round of review and you feel that this manuscript is now acceptable for publication, you may indicate that here to bypass the “Comments to the Author” section, enter your conflict of interest statement in the “Confidential to Editor” section, and submit your "Accept" recommendation.

Reviewer #1: All comments have been addressed

2. Is the manuscript technically sound, and do the data support the conclusions?

Reviewer #1: Yes

3. Has the statistical analysis been performed appropriately and rigorously? 

Reviewer #1: Yes

4. Have the authors made all data underlying the findings in their manuscript fully available?

Reviewer #1: Yes

5. Is the manuscript presented in an intelligible fashion and written in standard English?

Reviewer #1: Yes

6. Review Comments to the Author

Reviewer #1: (No Response)

7. PLOS authors have the option to publish the peer review history of their article (what does this mean?). If published, this will include your full peer review and any attached files.

Reviewer #1: No

---

## [Editor Report · Acceptance letter]

3 Sep 2020

PONE-D-20-02677R1 

An overseas business paradox: Are Japanese general contractors risk takers? 

Dear Dr. Kotani:

I'm pleased to inform you that your manuscript has been deemed suitable for publication in PLOS ONE. Congratulations! Your manuscript is now with our production department. 

Kind regards, 

on behalf of

Dr. Petre Caraiani 

Academic Editor

PLOS ONE